# Single and Binary Removals of Pb(II) and Cd(II) with Chemically Modified *Opuntia ficus indica* Cladodes

**DOI:** 10.3390/molecules28114451

**Published:** 2023-05-31

**Authors:** Carmencita Lavado-Meza, Miguel C. Fernandez-Pezua, Francisco Gamarra-Gómez, Elisban Sacari-Sacari, Julio Angeles-Suazo, Juan Z. Dávalos-Prado

**Affiliations:** 1Facultad de Ingeniería, Universidad Continental, Huancayo 12000, Peru; 2Escuela Profesional de Ingeniería Ambiental, Universidad Nacional Intercultural de la Selva Central Juan Santos Atahualpa, Chanchamayo 12856, Peru; 76828087@uniscjsa.edu.pe; 3Laboratorio de Nanotecnología, Facultad de Ingeniería, Universidad Nacional Jorge Basadre Grohmann, Tacna 23003, Peru; fgamarra@unjbg.edu.pe (F.G.-G.);; 4Facultad de Ingeniería Industrial, Universidad Tecnológica del Perú, Lima 15046, Peru; c21205@utp.edu.pe; 5Instituto de Química Física Rocasolano, CSIC, 28006 Madrid, Spain; jdavalos@iqfr.csic.es

**Keywords:** biosorption, *Opuntia ficus indica*, Pb(II) removal, Cd(II) removal, binary removal

## Abstract

In this study, cladodes of *Opuntia ficus indica* (OFIC), chemically modified with NaOH (OFICM), have been prepared, characterized, and tested as an effective biomass to remove Pb(II) and/or Cd(II) from aqueous media. At an optimum pH of 4.5, the adsorption capacity, q_e_, of treated OFICM was almost four times higher than that of untreated OFIC. The maximum adsorption capacities (q_max_) in the single removal of Pb(II) and Cd(II) were 116.8 and 64.7 mg g^−1^, respectively. These values were 12.1% and 70.6% higher than those for the corresponding q_max_ in binary removal, which indicates the strong inhibitive effect of Pb(II) on the co-cation Cd(II) in a binary system. Structural and morphological characterization have been carried out by FTIR, SEM/EDX, and point of zero charge (pH_PZC_) measurements. The SEM/EDX results confirmed that the metals are adsorbed on the surface. The presence of C-O, C=O, and COO- functional groups were identified by FTIR on both OFIC and OFICM surfaces. On the other hand, we found that the adsorption processes followed the pseudo-second-order kinetics for both single and binary systems, with a fast biosorption rate of Pb(II) and Cd(II). The equilibrium data (adsorption isotherms) were better described by Langmuir and modified-Langmuir models for single and binary systems, respectively. A good regeneration of OFICM was obtained with an eluent of 0.1 M HNO_3_. Therefore, OFICM can be efficiently reused to remove Pb or Cd, up to three times.

## 1. Introduction

Nowadays, due to the high industrialization degree in cities, large amounts of untreated wastewater are continuously discharged. These waters contain a wide variety of pollutants, such as heavy metals [1]. Several of these metals are extremely dangerous due to their non-biodegradability, persistence, high mobility, and significant accumulation capacity through the trophic chain [2].

Heavy metals such as Pb and Cd in polluted waters are noteworthy, particularly from untreated industrial waste related to mining, petrochemical, graphic printing, agricultural, electronic, coating, painting, or battery manufacturing/disposal activities [3]. In humans, chronic exposure to Pb(II) causes reproductive and neurological problems, as well as certain genotoxic and carcinogenic effects [4]. Similarly, chronic exposure to Cd(II) also produces lethal disorders, damage to kidneys, bones, and lungs, and erythrocytes’ destruction in blood [5].

The most common methods to remove heavy metals from wastewater are: ion exchange, chemical precipitation, reverse osmosis, ultrafiltration, and adsorption [6]. Among these methods, adsorption using biomasses (biosorption) stands out. Biosorption is an economical method, respectful to the environment, and is also easy to operate [7], since biomasses are biodegradable, available, and low-cost materials. For lead and/or cadmium removal, the literature reports a wide variety of biomasses, such as *Chrysopogon zizanioides* root powder [6], *Phyllanthus emblica* fruit stone [4], *Opuntia fuliginosa* and Agave angustifolia [8], *Opuntia ficus indica* [9,10], avocado pear [11], cucumber peal [12], rapeseed [13], taro [14], peanut shells [15], coffee and cocoa [16], etc. These biomasses include organic compounds, such as humic acids, lignin, cellulose, hemicellulose, proteins, etc., containing carbonyl, carboxyl, amine, and hydroxyl groups, which are adsorption sites capable of fixing heavy metals [17].

The *Opuntia ficus indica* (OFIC) is a cactus that belongs to the *Opuntiodeae* subfamily [18]. It is native to Mexico but has profusely developed in the central region of Peru, particularly at the Huamanga-Ayacucho [19]. It is a plant that does not require fertile soils, irrigation, or greater care [17], for which it has received attention not only as food but also for use as a biosorbent. According to previous studies [9,10], the OFIC biomass has functional groups such as hydroxyls, carboxylic, and carboxylates that would be associated with the adsorption of Pb and Cd. To improve the lead and/or cadmium removal capacities from aqueous solutions, the literature reports biomass modifications, particularly with alkaline solutions [20,21,22], managing to improve the biosorption capacity of the metal under study up to 55%, compared to the corresponding unmodified biomass.

Industrial effluents generally contain several metals (multi-metallic aqueous solutions), and the removal of one of them by adsorption is affected by the presence of the others (co-ions). Antagonistic effects are usually reported in the adsorption capacities of multi-metallic solutions [23,24,25,26], where different co-cations have different inhibitory effects, e.g., Cu(II) cannot be significantly affected; on the contrary, Ni(II) can be suppressed [27]. The decreased removal in multi-metallic systems is attributed to the competition among metallic ions for accessible binding sites on the biosorbents, and metal ion properties, such as electronegativity or the radius of metals, have an important role to explain these results [25,28]. Several authors have reported, in multi-metallic systems, that Pb(II) shows significant inhibitive effects on other co-ions, such as Cd(II), Zn(II), Hg(II), Cu(II), or Cr(III) [25,29,30,31,32].

In this work, we study the biosorption processes of Pb(II) and Cd(II) in both forms of removal, single and binary systems, through the *Opuntia ficus indica* biomass, treated with NaOH (OFICM). We show that this abundant and cheap biomass can be an efficient biosorbent to remove Pb and Cd from contaminated aqueous media, although it is important to take into account that the removal of Cd(II) is strongly affected by the presence of Pb(II).

## 2. Results and Discussion

### 2.1. pH_PZC_ Measurements: Effect of Alkaline Treatment

It is important to mention that after treatment with NaOH, the OFIC loses mass (almost 49%). This would be due to the dissolution of several organic structures of OFIC by the action of the alkaline treatment.

The point of zero-charge pH values (pH_PZC_) were obtained from the intersection of ΔpH (=pH_f_ − pH_0_) curves with the pH_0_-axis (see Figure 1): pH_PZC_ = 4.2 and 4.8, respectively, for OFIC and OFICM. Section 2.3 discusses the interpretation of this measure, studying the effect of pH on simple and binary adsorptions of Pb(II) and Cd(II).

Figure 2 depicts the significant increase of Pb(II) and Cd(II) adsorption capacities, q_e_, by the treated biomass, OFICM. Therefore, taking into account the optimal conditions described below, the q_e_, to remove Pb(II) and Cd(II), of OFICM was almost four times higher than that of OFIC (see Figure 2). Several authors have reported an increase in the metal removal capacity due to the alkaline treatment of biomasses [20,22,33,34,35].

### 2.2. FTIR Analysis: Morphological and Structural Characterization (SEM/EDX)

Figure 3 shows typical FTIR spectra of OFIC and OFICM, with similar absorption peaks or spectral bands, before and after biosorption of Pb(II) and Cd(II). The FTIR spectrum of OFIC (Figure 3a) shows typical bands of lignocellulosic compounds located at 3551.8–3048 cm^−1^, associated with axial deformations of O–H groups [17], at 2915.7 cm^−1^, attributable to the asymmetric vibration of C-H [8], at 1243.8 cm^−1^, associated with the presence of C-O bonds [36], at 1606.6 cm^−1^, associated with the stretching vibration of the double bond of C=O carboxylic groups, at 1384 cm^−1^, associated with the stretching vibration of –COO, and at 1032.8 cm^−1^, attributable to the vibration of -C-O-C- and -OH in polysaccharides [10].

The FTIR spectrum of OFICM (Figure 3b, in black) shows changes, concerning OFIC, in the intensity and position of some adsorption bands. Thus, the broad bands at positions 3551.8–3048 cm^−1^ and 1243.8 cm^−1^ practically disappeared, and the band at 1032.8 cm^−1^ became less intense. These results would indicate the structural modifications caused by the alkaline treatment of OFIC [22,27].

FTIR spectra of OFICM loaded with Pb(II) and Cd(II) (Figure 3b) show, as before the case, changes concerning clean OFICM in the intensity and position of some adsorption bands. Thus, the intensity of bands at 1606.6 and 1032.8 cm^−1^ decreased, and shifted towards higher frequencies: displacements at, respectively, Δ_1_ = 2.7 and Δ_2_ = 6.5 cm^−1^ for single adsorptions (OFICM + Pb, OFICM + Cd), and at Δ_1_ = 0.8 and Δ_2_ = 3.5 cm^−1^ for binary adsorption (OFICM + Pb + Cd). These displacements indicate a greater alteration of the bands associated with functional groups (OH, C-O-C, carboxylic) in single adsorptions than in binary ones.

Figure 4 shows the SEM morphologies and their corresponding EDX spectra for OFIC, OFICM, and loaded with Pb(II) and Cd(II). OFICM showed a more porous and cracked surface than OFIC, which would favor the adsorption of the metal [37].

Similar morphologies can be appreciated on the surfaces of (OFICM + Pb) and (OFICM + Cd), whereby both showed less porous areas than on the surfaces of OFICM. The morphology of (OFICM + Pb + Cd) was more homogeneous and compact than those separately loaded with the metals (single systems). EDX spectra of all samples showed common peaks associated with C, O, Ca, Al, and Si. Peaks associated with K, Mg, Cl, and S in OFIC disappeared in the OFICM spectrum. As expected, we can appreciate the presence of corresponding metal peaks in the EDX spectra of (OFICM + Pb), (OFICM + Cd), and (OFIM + Pb + Cd). It is interesting to note that if we take into account the intensity of the peak of O (1.2 KeV) as a reference, the ratios of Pb (2.4 KeV)/Cd (3.1 KeV) intensities were higher (almost three times) in binary removal than in simple ones. This result would be related to the higher affinity and strong inhibitory effect of Pb(II) on the co-cation Cd(II), described in the following sections.

### 2.3. Influence of pH Solution in the Single and Binary Adsorptions

The pH of the solution is an important factor in the biosorption of heavy metals since it significantly influences the adsorbent/adsorbate interaction [38,39]. Figure 5 shows the influence of pH on the Pb(II) and Cd(II) biosorption capacities, q_e_, of OFICM. Therefore, for acidic pH values, we can observe that the sorption capacity, q_e_, for both metals increased when increasing the pH, and reached a plateau around pH 4.5. In the same figure, the pH_PZC_ and the point at which Pb(II) starts rapid precipitation (s.p) have been marked. It is noticeable that maximum values of q_e_ were achieved at a pH close to 4.5, which is less than pH_PZC_ = 4.8. The interpretation of pH_PZC_ does not explain these results, since for pH < pH_PZC_, the OFICM surface would be positively charged and would not favor the electrostatic attraction of Pb(II) and Cd(II). Therefore, this mechanism could not be applied to the adsorption of Pb(II) and Cd(II) on OFICM. In this regard, various mechanisms for divalent metallic biosorption on low-cost biomasses have been proposed [40], e.g., (i) complexation reactions [41], (ii) ionic exchange [40,42], and (iii) microprecipitation [43].

### 2.4. Influence of OFICM Dose in Single and Binary Systems

Figure 6 shows the influence of the OFICM dose on the adsorption capacity, q_e_, of Pb(II) and Cd(II). For both systems, single and binary, we can appreciate a variation trend, similar among the q_e_ vs. OFICM dose curves. In the interval from 0.5 to 1 g L^−1^, the variation of q_e_ was smaller than in other measurement intervals. Therefore, we considered this interval as the most optimal for simple and binary removal of Pb(II) and Cd(II).

### 2.5. Influence of Initial Concentration of Pb(II) and/or Cd(II) Ions, C_0_

Figure 7 shows the effect of the initial metallic ion concentration, C_0_ (range from 10 to 200 mg L^−1^), on the adsorption capacity, q_e_, of OFICM, in single and binary systems. For both systems, it was observed that: (i) The adsorption capacity, q_e_, of Pb(II) and/or Cd(II) increased with C_0_. This behavior would be related to the driving force necessary to overcome the resistance to the mass transfer of metal ions from the solution to biosorbent [44]. (ii) Both the values and the growth trend of q_e_ were significantly higher for Pb(II) than for Cd(II), and in turn, higher in single than in binary systems. This last difference has also been observed in the (q_e_ vs. pH) and (q_e_ vs. dose) curves and would indicate the significant antagonistic effect of co-cations on the binary q_e_ of Pb(II) and Cd(II), which is discussed in the next sections.

### 2.6. Kinetic Data of Biosorption in Single and Binary Systems

The experimental kinetic data were adjusted using three kinetic models of adsorption: q_t_ vs. t for pseudo-first-order (Equation (1)) and pseudo-second-order (Equation (2)) models, and q_t_ vs. t^1/2^ for the intra-particle diffusion model (Equation (3)) [45]. The parameters obtained after the optimized adjustments and their corresponding qualities (correlation coefficient, R^2^) are reported in Table 1. Here, q_t_ has been defined as the adsorption capacity at time t, which is the amount of metal removed per unit mass of biosorbent at time t.
(1)$$q_{t,i}=\left(1-e^{k_{1}t}\right)$$
(2)$$q_{t,i}=\frac{{q_{e,i}}^{2}k_{2}\cdot t}{1+q_{e,i}\cdot k_{2}\cdot t}\;h=k_{2}q_{e,i}^{2}$$
(3)$$q_{t}=k_{\mathrm{di}}\cdot t^{1/2}+C$$
where, q_e_ is the metal adsorption capacity at the equilibrium (mg g^−1^), or merely the adsorption capacity, and $k_{1}$ is the kinetic constant (min^−1^) for the pseudo-first-order model. $k_{2}$ and h are, respectively, the rate constant adsorption (g mg^−1^ min^−1^) and the initial adsorption rate (mg g^−1^ min^−1/2^) for the pseudo-second-order model, and k_d_ (mg⋅g^−1^⋅min^−1/2^) is the intra-particle diffusion rate constant.

Figure 8 shows the results of the kinetic tests (q_t_ vs. t) carried out to determine the equilibrium time required for Pb(II) and Cd(II) adsorption on OFICM, in both single and binary systems. We can appreciate a rapid increase of Pb(II) and Cd(II) adsorption capacities, q_t_, within the first 10 min. The q_t_ equilibrium value (q_e_) was reached at approximately 120 min. For longer times (t > 120), q_t_ practically remained constant. For all cases, a better correlation (R^2^ ≥ 0.93) was obtained with the pseudo-second-order than with the pseudo-first-order adjustment model (see Table 1). Accordingly, we can affirm that: (i) the adsorption of Pb(II) and Cd(II) in both single and binary removals is a chemisorption process, and (ii) in the binary system, the calculated adsorption capacities, q_e_, of Pb(II) and Cd(II) were, respectively, ~12% and 61% lower than for single systems. The latter result indicates interference effects in the competitive binary removal of metals, being very affected by the Cd(II) adsorption.

The q_t_ vs. t^1/2^ plot (Figure 9) was fitted to the intra-particle diffusion Weber–Morris model. We can distinguish two well-defined parts: The first part (0 < t^1/2^ < 3) showed rapid growth of q_t_ at the time t (in min), which would indicate the rapid absorption of Pb(II) and/or Cd(II) ions on the outer surface of the biosorbent. The adsorption lines did not pass through the origin, which would indicate that the adsorption process in this region is not controlled only by the intra-particle diffusion model [46]. In the second part (t^1/2^ > 3), a slower growth of q_t_ with t was observed, which would be related to a gradual adsorption process, wherein Pb(II) and Cd(II) ions would enter and fill the biosorbent pores, until reaching equilibrium [11].

### 2.7. Adsorption Isotherms in Single and Binary Systems

Data obtained from Pb(II) and Cd(II) biosorption experiments (q_e_ vs. C_e_, Figure 10) were fitted to Langmuir, modified-Langmuir, and Freundlich equilibrium models. The isotherms were determined for optimal conditions at pH 4.5, room temperature (293 K), OFICM dose = 1 g L^−1^, contact time t = 120 min, and a range of equilibrium, C_e_, metal concentrations between 0 and 180 mg L^−1^. Figure 10 shows that all the fitting curves of the adsorption isotherms had a convex shape (type I isotherm), which, according to Moreira et al. [47], would be associated with the metallic adsorption by monomolecular layers of non-porous or microporous biosorbents.

Isotherm data of the single systems were fitted to the Langmuir (Equation (4)) and Freundlich (Equation (5)) models. The corresponding fitting parameters are consigned in Table 2.
(4)$$q_{e}=q_{\max}\frac{k_{L}\cdot C_{e}}{1+k_{L}C_{e}}$$
(5)qe,i=KFCe1n
where, $q_{e}$ is the equilibrium adsorption capacity of the biosorbent, C_e_ is the equilibrium metal concentration, $q_{\max}$ is the maximum equilibrium adsorption capacity (or merely the maximum adsorption capacity), k_L_ is the affinity constant between the adsorbate and the biosorbent, n is the constant related to the adsorption intensity of the biosorbent as a function of its degree of heterogeneity, and K_F_ is the Freundlich constant related to the adsorption capacity.

The adsorption isotherms for the single removal of Pb(II) and Cd(II) were well-fitted to the Langmuir model (R^2^ ≥ 0.91). The corresponding $q_{\max}$ calculated values were, respectively, 116.8 and 64.7 mg g^−1^.

Isotherm data of the binary systems (Figure 10b) were fitted to the extended Langmuir (ELM) (Equation (6)) and modified-Langmuir (MLM) (Equation (7)) models [47]. Both models assume complete competition of adsorption sites for different solutes. The corresponding fitting parameters have been consigned in Table 3.
(6)$$q_{\mathrm{ei}}=\frac{q_{\max,i}C_{\mathrm{ei}}k_{L,i}}{1+\sum_{j=1}^{N}k_{L,j}C_{e,j}}$$
(7)qei=qmax,ikL,iCeiηi1+∑j=1NkL,j(Cejηj)
where i and j are indices associated with Pb and Cd, respectively, q_max_ is the maximum adsorption capacity, k_L_ is the Langmuir affinity constant, and η is the correction factor.

The adsorption isotherms for the binary system were better fitted with the MLM (R^2^ = 1 and 0.98) than with the ELM (R^2^ = 0.91 and 0.97). From the MLM (Table 2), we can derive the maximum sorption capacity, q_max_, equal to 102.7 and 19.0 mg g^−1^, to remove Pb(II) and Cd(II), respectively. These values are lower than in the corresponding single systems, particularly in the binary removal of Cd(II), which was 70.6% lower than in the single removal, while for Pb(II), it was 12.1% lower. This result shows that Pb(II) caused a strong antagonistic effect on the removal of the co-ion Cd(II) in binary systems. Several authors have also reported the strong inhibitive effect of Pb(II) on Cd(II), in both binary and ternary systems [11,12,25,28,29,31].

The greater competitiveness of Pb(II) compared to Cd(II) ions, to occupy the accessible binding sites in OFICM, would be related to several factors, such as specific binding site structures, chemistry of the solution, adsorbate hydration ionic properties, etc. Although it is not easy to determine a common factor [48], it is important to highlight the explanation that takes into account the higher electronegativity (2.33) and the smaller hydrated ionic radius (410 pm) of lead, compared to cadmium (1.7 and 426 pm), as a possible factor that would favor the greater biosorption of Pb(II) than Cd(II) [12,29].

Our results indicate that Pb(II) can be even effectively removed by OFICM from aqueous solution in the presence of Cd(II), but the removal of this cation is strongly affected by the presence of Pb(II).

The maximum Pb(II) and/or Cd(II) biosorption capacities, q_max_, of biomasses similar to that studied in this work are shown in Table 3. It is interesting to note that the q_max_ of OFICM was among the highest, particularly for Cd(II) removal, which, as expected, was almost twice the corresponding untreated biomass [10]. Even though in a binary system the removal of Cd(II), using OFICM, is significantly low, the value obtained here was one of the highest, compared to those reported in the literature.

**Table 3 molecules-28-04451-t003:** Comparative table of the maximum adsorption capacities, q_max_, of treated biomasses, to remove Pb(II) and Cd(II).

Biomass	q_max_ (mg g^−1^)		Reference
	Pb(II)	Cd(II)	
Olive Stone ^a^	15	-	[49]
	≤25.48	-	[50]
Rice bran ^a^	78.9	-	[51]
*Moringa oleifera* tree leaves ^a^	209.55	-	[52]
*Auricularia auricular* ^a^	36.35	-	[37]
*Arabica*-coffee ^a^	223.1	-	[33]
*Theobroma*-cocoa ^a^	303.0	-	[33]
Apricot shells ^a^	37.37	-	[21]
*Mangifera indica* seed shells ^a^	59.25	-	[36]
*Coconut Shaft*	22.1		[39]
*Nostoc commune* ^a^	384.6		[53]
Algae waste biomass ^a^	-	53.19	[38]
*Mauritia flexuosa* ^a^	202.43 (56.48) ^b^	178.94 (22.05) ^b^	[54]
Carob shells ^a^	-	49.63	[35]
Alga (*Hydrodictyon reticulatum*) ^a^	-	12.74	[34]
Soy waste biomass ^a^	82.8	46.08	[22]
Water hyacinth, *Eichhornia crassipes* ^c^	26.32 (25.38) ^b^	12.59 (4.05) ^b^	[31]
*Opuntia ficus indica* cladodes ^a^	116.8 (102.66) ^b^	64.7 (19.03) ^b^	This work

^a^ NaOH treatment. ^b^ Binary removal. ^c^ Acid treatment.

### 2.8. Biosorption Thermodynamics

The thermodynamic functions, standard Gibbs energy (∆G^0^), enthalpy (∆H^0^), and entropy changes (∆S^0^) for the biosorption processes, in both single and binary systems, were evaluated using the Equations (8) and (9) (Van’t Hoff equation):(8)$$\Delta G^{0}=-\mathrm{RT}\ln\left(K_{c}\right)$$
(9)$${\ln\;K}_{c}=\frac{\Delta S^{0}}{R}-\frac{\Delta H^{0}}{\mathrm{RT}}$$
where K_c_ is the equilibrium constant, $K_{c}=\frac{C_{\mathrm{es}}}{C_{e}}$, and C_es_ and C_e_ are the equilibrium Pb(II) and/or Cd(II) concentrations, respectively, in the biosorbent and in the solution. R is the universal gas constant and T is the temperature of the solution. The thermodynamic function values obtained are consigned in Table 4. For both single and binary systems, the negative values of ΔG^0^ indicate that the Pb(II) and Cd(II) adsorption processes on OFICM are spontaneous and favorable for all the considered temperatures (Table 4). The positive values of the entropy change, ΔS^0^, evidenced the increasing randomness between the solid/solution interface during biosorption. On the other hand, the positive values of ΔH^0^ indicated that the studied processes are endothermic. Similar results were seen in the single removal of Pb(II) and Cd(II) with microalgae *Neochloris oleoabundans* [25].

### 2.9. Desorption and Regeneration of OFICM

Biosorbents, coming from abundant biomasses and with high adsorption/desorption capacities, are interesting materials (e.g., to remove heavy metals) with economic and environmental viability [53]. We have studied the adsorption/desorption capacities of OFICM to find its effective regeneration and stability. In Figure 11, the results of the Pb(II) and Cd(II) desorption efficiency (%D, calculated using Equation (11)) experiments using four acidic eluents: 0.1 M of HNO_3_, HSO_4_, HCl, and CH_3_COOH, are presented. Acids were used because these release H^+^ protons, which are replaced by metallic cations such as Pb(II) and Cd(II) on the biosorbent surface [45].

Except for CH_3_COOH, the other acids showed a high desorption efficiency (%D > 70), particularly for Pb(II), with HNO_3_ being the most efficient eluent with which Pb(II) and Cd(II) were desorbed up to %D = 96 and 88, respectively. With this eluent, up to four adsorption/desorption cycles were carried out (Figure 12). For both Pb(II) and Cd(II) adsorbates, the recovery efficiency was considerable (>50%) up to three cycles, and after that, the efficiency significantly decreased (%D < 30).

## 3. Materials and Methods

### 3.1. Preparation of Opuntia ficus indica

Untreated biomass (OFIC): *Opuntia ficus indica* cladodes were collected from the Pilcomayo District, located in the Junín Region of Peru. The samples were previously washed with drinking water and then rinsed with distilled water. Once the spines were removed and cut into small pieces, the cladodes were dried for 72 h in an oven at 333 K, and finally, ground and sieved.

Treated biomass (OFICM): OFIC powder was mixed with 0.1 M NaOH solution in a ratio of 1 g:10 mL for 24 h, at room temperature (293 K) with constant stirring (500 rpm). Afterward, the mixture was filtered and washed with abundant deionized water until reaching a colorless solution and a constant pH. Finally, the sample was dried at 313 K, then it was ground, and sieved with a mesh < 0.3 mm.

### 3.2. Preparation of Pb and Cd Solutions

Solutions at different concentrations of Pb and Cd were prepared by diluting appropriate amounts of stock solutions, which were obtained from Pb(NO_3_)_2_ and Cd(NO_3_)_2_·4H_2_O salts. All chemical reagents used in this work were of analytical grade.

### 3.3. Bisorbent Characterization

The point of zero-charge pH values (pH_PZC_) were determined according to the procedures reported by do Nascimento et al. [55]. Mixtures of 0.05 g of biosorbent with 50 mL of aqueous solutions were prepared under different initial pH values (pH_0_), ranging from 1 to 12. The acid solutions were prepared from 1 M HCl, while the basic solutions from 1 M NaOH. After 24 h of equilibrium, the final pH values (pH_f_) were determined.

A Fourier transform infrared spectrophotometer (FTIR, Bruker Invenio R/Platium ATR) was used to identify the functional groups present on the surface of the biosorbents. The wavelength was set to 4000 to 400 cm^−1^.

Morphological and elemental analysis of the biosorbent surface was performed by scanning electron microscopy (SEM) coupled with EDX (energy-dispersive X-ray spectroscopy) (Thermo Scientific Co., Eindhoven, Netherlands).

### 3.4. Biosorption Assays

Batch experiments, in both single and binary systems, were carried out by mixing 25 mg of OFICM with 25 mL of the solution containing Pb(II) and/or Cd(II) at different initial concentrations, C_0_ (from 10 to 200 mg L^−1^). These solutions were adjusted to pH in the range from 2.0 to 7.0 by adding appropriate amounts of 0.1 M HNO_3_ or 0.1 M NaOH. The suspension was stirred to 300 rpm, at room temperature (293 K), for the period from 0 to 120 min. For all cases, after biosorption, the phases were separated by filtration.

The Pb(II) and Cd(II) concentrations, before and after adsorption, were evaluated using an Atomic Absorption Spectrophotometer (SHIMADZU-AAS 6800, Kyoto, Japan). All adsorption experiments were replicated three times, and the results were averaged.

The equilibrium adsorption capacity (or merely the adsorption capacity) of OFICM was determined by Equation (10):(10)$$q_{e,i}=\frac{\left(C_{0,i}-C_{e,i}\right)V}{m}$$
where i is the index associated with Pb or Cd. C_0,i_ and C_e,i_ (in mg·L^−1^) are the initial and equilibrium concentrations of the metallic ion, respectively, V (in L) is the volume of the solution, and m (in g) is the biosorbent mass.

The desorption process was carried out using four types of acidic eluents: 0.1 M of HNO_3_, HSO_4_, HCl, and CH_3_COOH. A total of 50 mg of OFICM biosorbent previously loaded with 1.25 mg of Pb(II) or Cd(II) (C_0_ = 25 mg L^−1^) was subjected to the desorption process, by adding 50 mL of each eluent and then stirring at 300 rpm for 3 h. After that, the biosorbent was washed with distilled water, dried, and reused again. The adsorption/desorption operation was repeated up to 4 times. The adsorbed and desorbed metal concentrations were analyzed by atomic absorption spectroscopy (previously described). The desorption efficiency percentage (%D) of OFICM was calculated using the following expression [56]:(11)$$\%D=\frac{\mathrm{Amount}\;\mathrm{of}\;\mathrm{metal}\;\mathrm{ion}\;\mathrm{desorbed}}{A\mathrm{mount}\;\mathrm{of}\;\mathrm{metal}\;\mathrm{ions}\;\mathrm{adsorbed}}\times 100$$

## 4. Conclusions

*Opuntia ficus indica* cladodes (OFIC) were chemically modified with 0.1 M NaOH, to improve their Pb(II) and Cd(II) adsorption capacities, in both single and binary systems. In single systems under similar conditions, the adsorption capacities, q_e_, to remove Pb(II) or Cd(II), on the treated biosorbent (OFICM), were (q_e,Pb_ = 47.8, q_e,Cd_ = 30.6 mg g^−1^) almost four times higher than the corresponding q_e_ (q_e,Pb_ = 13.6, q_e,Cd_ = 7.3 mg g^−1^) on the untreated biomass (OFIC). Therefore, the alkaline treatment increased the basic or active sites on OFICM.

The point of zero charge, pH_PZC_, of OFICM (4.8) was greater than that of OFIC (4.2). Considering that the optimal adsorption capacities, q_e_, were determined at pH < pH_PZC_, the interpretation of pH_PZC_ could not be applied to the adsorption of Pb(II) and Cd(II) on OFICM.

The active sites on OFICM were associated with OH, C=O, and –COO functional groups, identified by FTIR, whose absorption bands showed greater alterations in single than in binary systems. The SEM/EDX analyses of OFICM showed a more porous and cracked surface than those of OFIC. On the other hand, the morphology of (OFICM + Pb + Cd) was more homogeneous and compact than that of (OFICM + Pb) and (OFICM + Cd). These results are consistent with the higher affinity and strong inhibitory effect of Pb(II) on the co-cation Cd(II).

For both single and binary systems: (i) The adsorption capacities, q_e_, to remove Pb(II) and Cd(II), increased with the initial concentration of the corresponding metals, C_0_. Both the values and the growth trend of q_e_ were significantly higher for Pb(II) than for Cd(II) removal. (ii) The corresponding adsorption kinetic data were well-fitted with the pseudo-second-order model, indicating that the Pb(II) and Cd(II) adsorptions on OFICM were chemisorption processes. (iii) The intra-particle diffusion Weber–Morris model indicated rapid adsorption of Pb(II) and Cd(II) in the first stages of the biosorption process.

At optimum conditions (pH 4.5, dose = 1 g L^−1^, T = 293 K, t = 120 min), the adsorption isotherms’ data (q_e_ vs. C_e_) were well-fitted with the Langmuir model for single systems and the modified-Langmuir model for binary systems. From these models, the maximum adsorption capacities were derived: In the single removal, q_max_ was equal to 116.8 and 64.7 mg g^−1^ for Pb(II) and Cd(II), respectively. In binary removal, the corresponding values decreased by almost 12% and 71%. These results indicate the strong inhibitive effect of Pb(II) on the co-cation Cd(II) in binary systems. Therefore, Pb(II) can be effectively removed from the aqueous solution in the presence of Cd(II), but the removal of this cation is strongly affected by the presence of Pb(II).

The q_max_ values determined in this work for the removal of Pb and Cd were higher than other similar biosorbents reported in the literature.

The biosorption process of Pb(II) and Cd(II) on OFICM, in both single and binary removal, can be characterized as endothermic (ΔH^0^ > 0), feasible and spontaneous (ΔG^0^ < 0), and with increasing randomness (ΔS^0^ > 0) at the solid/liquid interface.

Desorption (with HNO_3_)–regeneration experiments also showed that OFICM can be an efficient biosorbent to remove Pb and/or Cd. Therefore, OFICM can be reused up to three times with acceptable efficiency (%D > 50).

## Figures and Tables

**Figure 1 molecules-28-04451-f001:**
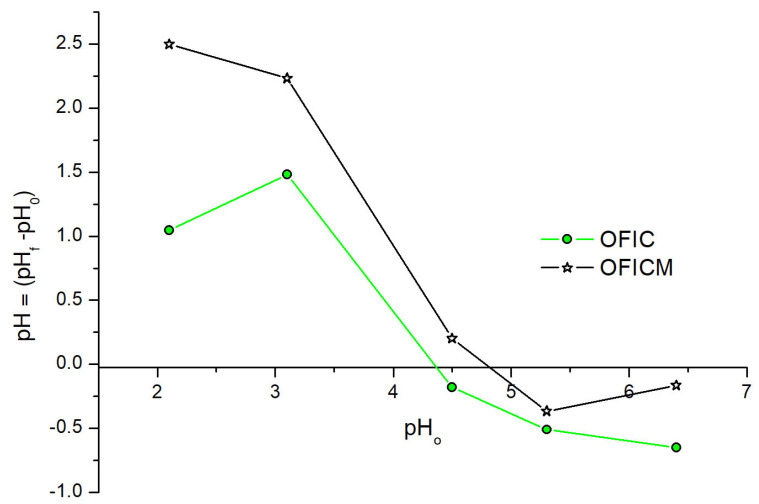
Determination of pH_PZC_ values for OFIC and OFICM biosorbents.

**Figure 2 molecules-28-04451-f002:**
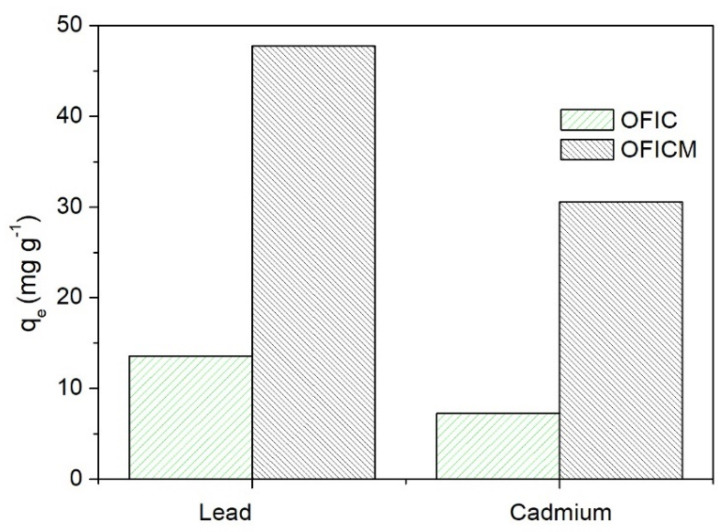
Adsorption capacity, q_e_, of *Opuntia ficus indica*, untreated (OFIC, green) and treated with NaOH (OFICM, dark). Initial Pb(II) concentration, C_0_ = 50 mg L^−1^, biosorbent dose = 1 g L^−1^, pH 4.5, contact time t = 120 min.

**Figure 3 molecules-28-04451-f003:**
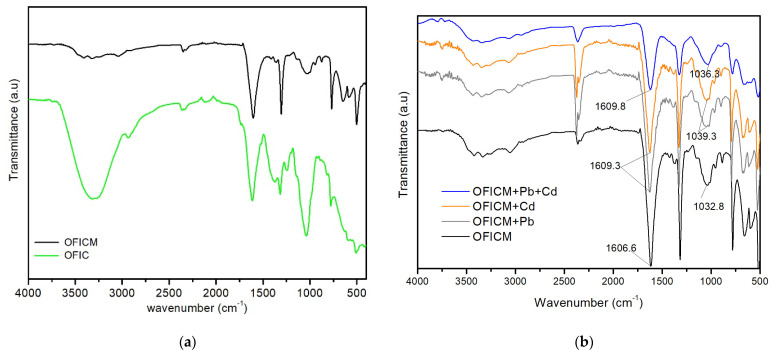
FTIR spectra of untreated (OFIC) and treated (OFICM) biosorbents: (**a**) before and (**b**) after Pb(II) and Cd(II) adsorptions.

**Figure 4 molecules-28-04451-f004:**
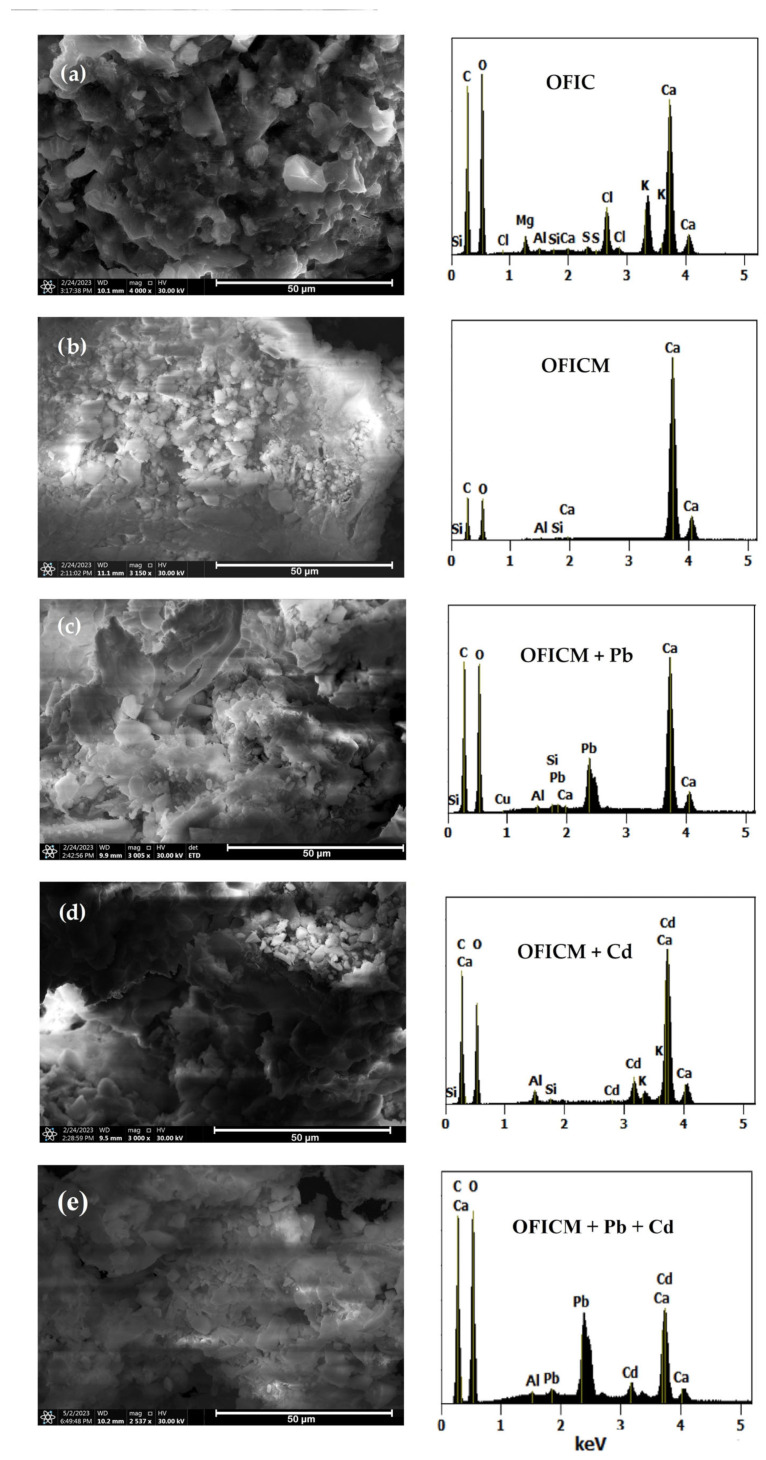
SEM images and EDX spectra of (**a**) OFIC, (**b**) OFICM, (**c**) OFICM + Pb, (**d**) OFICM + Cd, and (**e**) OFIDM + Pb + Cd. pH 4.5, C_0_ = 100 mg L^−1^, T = 293 K, t = 120 min.

**Figure 5 molecules-28-04451-f005:**
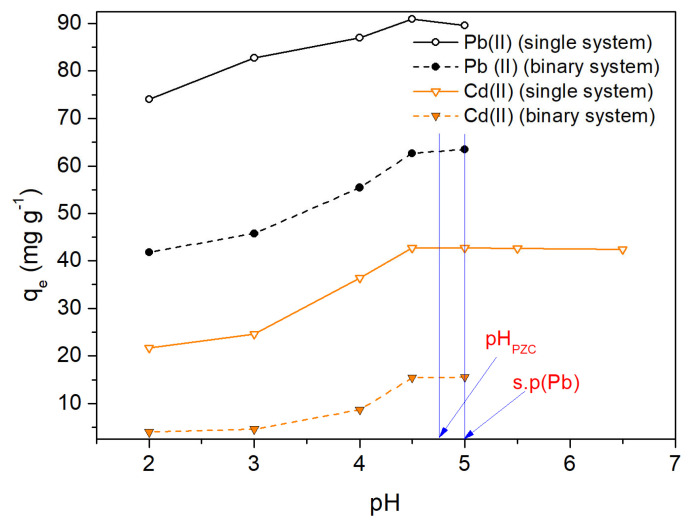
Influence of pH on the Pb(II) and Cd(II) adsorption capacity, q_e_, in single and binary systems. Experimental conditions: OFICM dose = 1 g L^−1^, T = 298 K, t = 120 min, C_0_ = 100 mg L^−1^.

**Figure 6 molecules-28-04451-f006:**
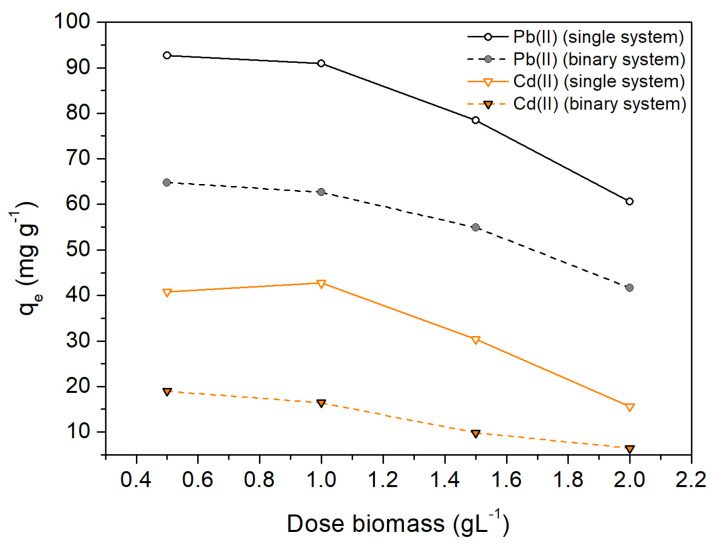
Influence of the dose of biomass on the removal capacity, q_e_. Experimental conditions: T = 298 K, t = 120 min, C_0_ = 100 mg L^−1^, pH = 4.5.

**Figure 7 molecules-28-04451-f007:**
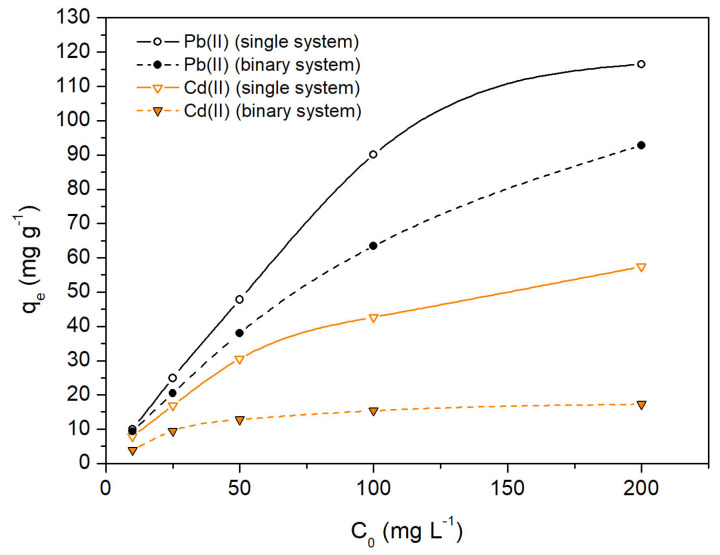
Influence of the initial concentration of metal ions (C_0_) on the q_e_ of OFICM. Experimental conditions: pH = 4.5, t = 120 min, T = 293 K.

**Figure 8 molecules-28-04451-f008:**
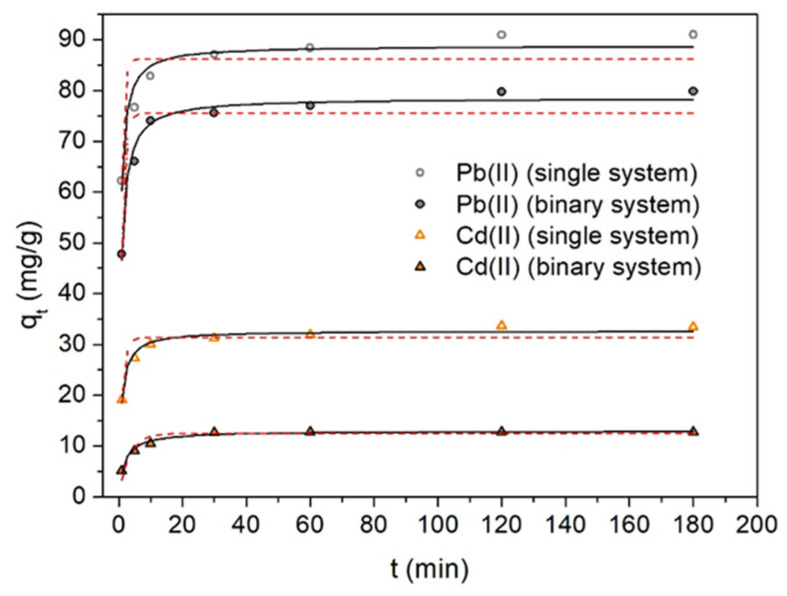
q_t_ (amount of Pb(II) and/or Cd(II) removed per mass unit of biosorbent) vs. contact time t (min). pH 4.5, T = 293 K, OFICM dose = 1 g L^−1^, C_0_ = 100 mg L^−1^. Pseudo-first-order (solid line) and pseudo-second-order (dotted lines) adjustments.

**Figure 9 molecules-28-04451-f009:**
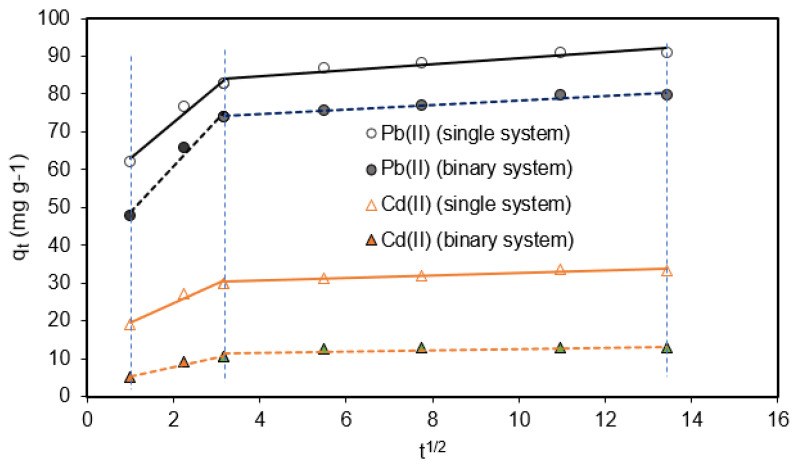
Intra-particle diffusion Weber–Morris plots of Pb(II) and Cd(II) adsorption on OFICM biosorbent. t in min, pH 4.5, T = 293 K, OFICM dose = 1 g L^−1^, C_0_ = 100 mg L^−1^. Pseudo-first-order (solid line) and pseudo-second-order (dotted lines) adjustments.

**Figure 10 molecules-28-04451-f010:**
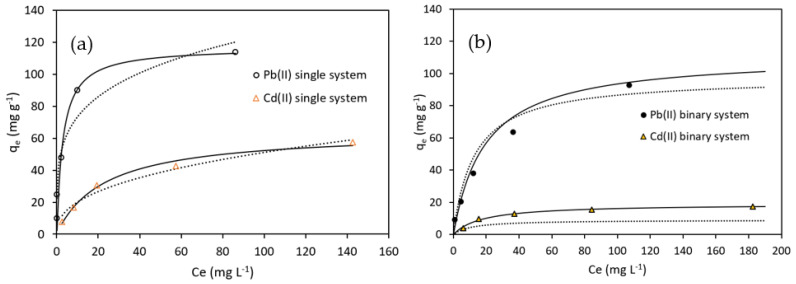
q_e_ vs. C_e_ adsorption isotherms. (**a**) Single-systems, Langmuir (solid lines), and Freundlich (dotted lines) adjustment models. (**b**) Binary systems, modified-Langmuir (solid lines), and extended Langmuir (dotted lines) adjustment models. OFICM dose = 1 g L^−1^, T = 293 K, t = 120 min, pH 4.5.

**Figure 11 molecules-28-04451-f011:**
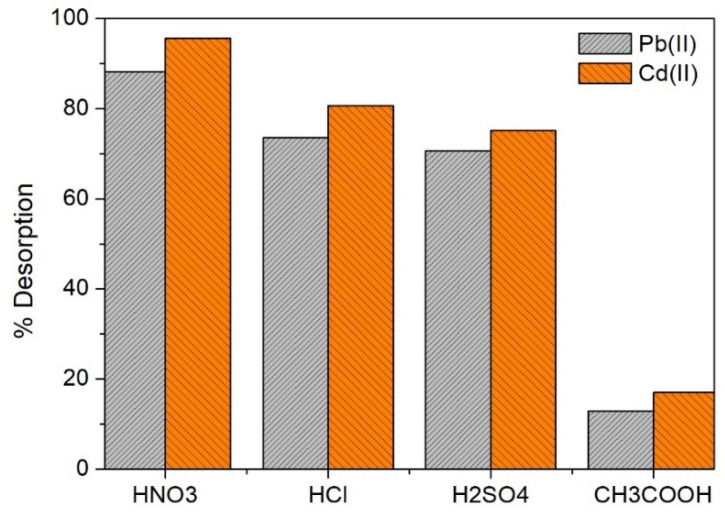
Desorption efficiency (%D) of OFICM using four different acidic eluents. C_0_ (Pb(II) or Cd(II)) = 25 mg L^−1^.

**Figure 12 molecules-28-04451-f012:**
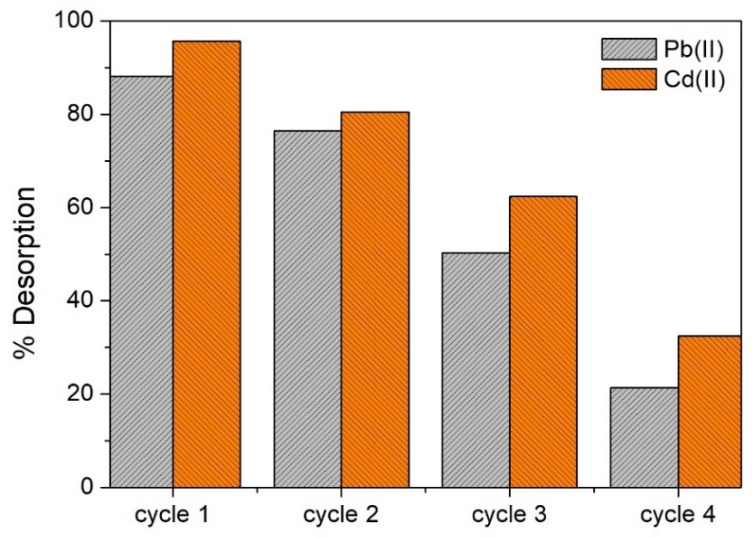
Desorption efficiency (%D) of OFICM vs. metal adsorption/desorption cycles. Eluent = 0.1 M HNO_3_. OFICM dose = 1 g L^−1^.

**Table 1 molecules-28-04451-t001:** Kinetic parameters for the adjustment of experimental data using kinetic models. pH 4.5, T = 293 K, OFICM dose = 1 g L^−1^, C_0_ = 100 mg L^−1^.

Model/Parameter	Single System		Binary System	
	Lead	Cadmium	Lead	Cadmium
Pseudo-first-order				
q_e_ (mg g^−1^)	86.2	31.4	75.6	12.4
k_1_ (min^−1^)	1.27	0.89	0.96	0.31
R^2^	0.71	0.80	0.81	0.85
Pseudo-second-order				
q_e_	88.9	32.7	78.6	13.0
k_2_	0.023	0.040	0.019	0.043
h	181.6	42.8	117.2	7.2
R^2^	0.93	0.97	0.97	0.97
Intra-particle diffusion				
k_d,I_	9.64	5.14	12.27	2.49
R^2^	0.98	0.96	0.98	0.97
k_d,II_	0.77	0.35	0.61	0.17
R^2^	0.89	0.92	0.96	0.85

k_1_ (min^−1^): the first-order kinetic constant; q_e_ (mg⋅g^−1^): calculated adsorption capacity; k_2_ (g⋅mg^−1^⋅min^−1^): rate constant adsorption; h (mg⋅g^−1^⋅min^−1^): initial adsorption rate; k_d_ (mg⋅g^−1^⋅min^−1/2^): intra-particle diffusion rate constant.

**Table 2 molecules-28-04451-t002:** Adjustment parameters of Pb(II) and Cd(II) biosorption isotherms, in single and binary systems.

Model/Parameter	Single System		Binary System	
	Pb(II)	Cd(II)	Pb(II)	Cd(II)
Langmuir				
q_max_ (mg g^−1^)	116.8	64.7	-	-
k_L_ (L mg^−1^)	0.35	0.04	-	-
R^2^	0.91	0.98	-	-
Freundlich				
K_F_ (mg g^−1^ L^(1/n)^ mg^−(1/n)^)	43.22	8.00	-	-
n_F_	4.35	23.34	-	-
R^2^	0.90	0.96	-	-
Modified-Langmuir (MLM)				
q_max_	-	-	102.7	19.0
k_L_	-	-	0.049	0.045
η	-	-	0.86	0.96
R^2^	-	-	0.98	1
Extended Langmuir (ELM)				
q_max,_	-	-	96.8	12.2
k_L_	-	-	0.07	0.02
R^2^	-	-	0.97	0.91

**Table 4 molecules-28-04451-t004:** Thermodynamic parameters of biosorption of Pb(II) and Cd(II) on OFICM, in single and binary systems.

	∆H^0^ (kJ mol^−1^)	∆S^0^ (J mol^−1^ K^−1^)	∆G^0^ (kJ mol^−1^)		
			293 K	303 K	313 K
Single					
Pb	52.36	197.61	−5.62	−7.19	−11.13
Cd	17.06	55.57	−0.31	−0.71	−0.89
Binary					
Pb	17.06	62.60	−1.26	−1.93	−3.05
Cd	17.05	46.52	−2.61	−3.36	−4.14

## Data Availability

Not applicable.

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
