# Peer review of "Single and Binary Removals of Pb(II) and Cd(II) with Chemically Modified Opuntia ficus indica Cladodes"

_molecules, 2023, doi:10.3390/molecules28114451_

Round 1
Reviewer 1 Report
Report on the manuscript ID: molecules-2366706 entitled: “Single and binary removals of Pb(II) and Cd(II) with Opuntia ficus indica cladodes chemically modified.”
Authors: Carmencita Lavado-Meza *, Miguel C. Fernandez-Pezua, Francisco Gamarra-López, Elisban Sacari-Sacari, Julio Angeles-Suazo, Juan Z. Dávalos-Prado
General aspects:
This manuscript addresses the removal of Pb(II) and Cd(II) in aqueous solution by using chemically modified Cladodes of Opuntia (OFIC). In a first step, OFIC which is a biomass was prepared by a treatment with NaOH to provide the modified biomass OFICM. In a second step the as-modified biomass was used as biosorbent for the removal of Pb(II) and Cd(II) in both single and binary adsorption removals. Several physico-chemical techniques, including pH at the point of zero charge, FTIR, and SEM couple with EDX, were used to characterize the biosorbent.
In general, we found that the literature review was poor and didn’t highlight the main problem which is supposed to be solved in the present paper. Indeed, few references [20-23] were given as previous investigations dealing with Pb(II) and Cd(II) removal by using modified biomass, the removal capacities being improved by a treatment with alkaline solution. However, the introduction did not provide the main research question in the removal of the two cations. What was the main problem in the removal of Pb(II) and Cd(II) by using biomass modification as described by the previous investigations? Were there any limitations in the previous investigations so that it justifies the present study? The fact that previous investigations only collected information on the binary or simultaneous removal of the two metals doesn’t constitute in itself a limitation which justifies that the present investigations could be considered as a novelty. Following the literature review (see for example reference [10]), it was reported that the decrease of the fixation of lead for greater pH was due to the complexation of lead ions by OH- groups which would prevent the metal biosorption. Another fact is that whereas Pb(II) is adsorbed as hydrolyzed species, this is not the case for Cd(II), a behavior which was linked to several factors explaining why Pb(II) was supposed to be likely adsorbed through inner-sphere surface complexations. These are for example some limitations which were highlighted in the simultaneous removal of the metals. As a matter of fact, the authors didn’t even mention these limitations in their literature review and why the alternative of carrying out single removal could be better taking into account all these limitations. We think that the novelty should rely on basic problems which occur in the simultaneous removal, for example the type of interference in the adsorption of the two metals, and how the authors hope to improve or solve this problem.
We think that the manuscript introduction lacks of some relevant references supporting the present work, and the justifications why they have undertaken the present investigations, taken into account previous results in the literature in terms of:
- What is (are) the problem(s)? What is the main research question in the removal of the two metals?
- Are there any existing solutions?
- Which one is the best?
- What is (are) the main limitation(s) regarding the literature review?
- How do you hope to improve or contribute to this?
Moreover, we could not see in the manuscript some basic experiences such as the effect of adsorbent dosage and the effect of temperatures (thermodynamic studies).
Please revise the presentation of the manuscript by using the classic IMRAD (Introduction, Methods, Results and Discussion) format.
Specific aspects:
Please find in the following lines some specific comments.
1. Introduction
Revise the introduction as noted in “General aspects”, by highlighting the specific research question and the basic limitations as given in previous literature.
2. Results and discussion
1) Please describe the bands positions with the wavenumber instead of using the positions 1, 2, etc.
2) Moreover, to better appreciate the displacement, we advise the authors to assign the wavenumber to the corresponding bands which are in the figure.
3) Please write Figure 3a and Figure 3b instead of Fig 3 left and Fig. 3 right.
4) Figure 3 represents the FTIR spectra of treated OFICM after Pb-loaded or Cd-loaded. This figure corresponds to single removal of Pb(II) or Cd(II). Since the authors are investigating both forms of removal, single and binary removals, they should compare the FTIR spectra of OFICM in both forms of removal. In other word, they should represent the FTIR spectra of OFICM + Pb + Cd and compare the spectra with the previous ones related to the single removals (OFICM + Pb, OFICM + Cd). This comparison could help in the interpretation of the type of interference which occur in the adsorption process.
5) Idem for Figure 4 with the SEM images: SEM images of OFICM + Pb or OFICM + Cd to be compared with the SEM images of OFICM + Pb + Cd.
6) What can we conclude after analyzing the results obtained in 4) and 5)?
7) Influence of pH solution: Figure 5
Please revise the unit of the ordinate axis: the unit of
must be in mg/g and not mg/L
The pHPZC was determined to be 4.8 according to Figure 1. As a result:
· for pH < 4.8, OFICM surface is positively charged,
· for pH > 4.8, OFCIM surface is negatively charged.
The authors investigated the influence of the pH in the range from 2 to 7.0. According to the concept of pHPZC, since the OFICM surface is positively charged in the pH range from 2.0 to 4.8 and Pb(II) being a positive free ion, a significant adsorption is not expected. However, in figure 5 for Pb(II) single removal, we have already at a pH value of 2, a significant adsorption of Pb(II) (which is greater than 70 mg/g). This is really contradictory to the concept of pHPZC.
We encourage the authors to refer to the Pourbaix-Diagram of Pb and Cd in order to state the exact state of the two ions and explain what exactly happens. Since electrostatic interactions are not favored, there must be other processes which could explain why at these values, the authors obtained such a high value of
.
As a matter of fact, since the pHPZC value is 4.8, what could explain that at a pH value of 4.5 (a value smaller than the 4.8), a maximum value is reached for
? At this value, the surface is still positively charged and Cd(II) and Pb(II) are also positively charged and are not expected to significantly interact with a positive surface charge. This is contradictory with the concept of pHPZC. The authors should really revise their interpretation.
In this paper, the authors based their analysis on both forms of removal, single and removal, through OFICM. It is then expected that all results which are given in the paper highlight what occurs in both single and binary removal. In figure 5, the influence of the pH solution was carried out only for the single removal of Pb(II) or Cd(II). In figure 6 for example, the influence of initial metal concentration was carried out for both single and binary removal. It is important to follow the same logic as in figure 6 and to compare the results obtained in both systems. What would be the influence of pH if the removal were carried out following the binary system? As a matter of fact, such binary removal was carried out in the paper of reference [10]. We advise the authors to do the same and to compare the results for both single and binary removal.
8) Influence of initial concentration
Lines 161-163: “It is also important to note that the qe values of Pb(II) and Cd(II) in a binary system are lower, particularly for Cd(II), than the corresponding values in a single system. These results indicate that the presence of both metals produce mutual interference effects on their adsorption processes.”
Question: What type of mutual interference effects? Are these effects supported by relevant literature of experience? It would have been better to carry out additional experiences (FTIR, XRD…) to exactly state what is really occurring in this binary system?
9) As already said, the authors should carry out the following experiences: the effect of the adsorbent dosage, effect of temperature (thermodynamic studies). See for example references [10, 23] where these experiences have been carried out.
10) Comparative table
In the comparative table, comparison has been made with mixed systems: those which are treated with NaOH and those which are not. We think that such comparison can only be made with similar systems, we mean systems which have been chemically treated. This table must be revised and compare only systems which have been treated. Untreated OFIC could not be compared with treated OFIC. As a matter of fact, we refer to the investigations which have been carried out in reference [10] by using untreated OFIC (Opuntia ficus indica cladodes). Although the OFIC has not been treated, significant amount of qe was reached: 98.62 mg.g-1 and 30.42 mg.g-1 for Pb(II) and Cd(II) respectively, in a binary removal.
11) Since the authors have conducted single removal, the question arises as to Pb2+ adsorption mechanisms when using OFICM: surface complexation, ion exchange mechanism, etc…However, nothing was said on such possible mechanisms.
12) Miscellaneous
The whole paper contains some sentences which must be corrected (grammar, vocabulary).
What is the difference between lines “256-260” and lines “261-265”?
It seems to be the same sentences.
General conclusion of the report:
The manuscript must be deeply revised and we recommend to carry out additional experiences. The authors should define the main research problem, recall past results and their limitations. From these limitations, they should define what they intend to do to solve the research question.
The whole paper contains some sentences which must be corrected (grammar, vocabulary).
Author Response
RESPONSE TO REVIEWER COMMENTS
- Regarding to Reviewer #1:
Comment: This manuscript addresses the removal of Pb(II) and Cd(II) in aqueous solution by using chemically modified Cladodes of Opuntia (OFIC). In a first step, OFIC which is a biomass was prepared by a treatment with NaOH to provide the modified biomass OFICM. In a second step the as-modified biomass was used as biosorbent for the removal of Pb(II) and Cd(II) in both single and binary adsorption removals. Several physico-chemical techniques, including pH at the point of zero charge, FTIR, and SEM couple with EDX, were used to characterize the biosorbent.
In general, we found that the literature review was poor and didn’t highlight the main problem which is supposed to be solved in the present paper. Indeed, few references [20-23] were given as previous investigations dealing with Pb(II) and Cd(II) removal by using modified biomass, the removal capacities being improved by a treatment with alkaline solution. However, the introduction did not provide the main research question in the removal of the two cations. What was the main problem in the removal of Pb(II) and Cd(II) by using biomass modification as described by the previous investigations? Were there any limitations in the previous investigations so that it justifies the present study? The fact that previous investigations only collected information on the binary or simultaneous removal of the two metals doesn’t constitute in itself a limitation which justifies that the present investigations could be considered as a novelty. Following the literature review (see for example reference [10]), it was reported that the decrease of the fixation of lead for greater pH was due to the complexation of lead ions by OH- groups which would prevent the metal biosorption. Another fact is that whereas Pb(II) is adsorbed as hydrolyzed species, this is not the case for Cd(II), a behavior which was linked to several factors explaining why Pb(II) was supposed to be likely adsorbed through inner-sphere surface complexations. These are for example some limitations which were highlighted in the simultaneous removal of the metals. As a matter of fact, the authors didn’t even mention these limitations in their literature review and why the alternative of carrying out single removal could be better taking into account all these limitations. We think that the novelty should rely on basic problems which occur in the simultaneous removal, for example the type of interference in the adsorption of the two metals, and how the authors hope to improve or solve this problem.
We think that the manuscript introduction lacks of some relevant references supporting the present work, and the justifications why they have undertaken the present investigations, taken into account previous results in the literature in terms of:
- What is (are) the problem(s)? What is the main research question in the removal of the two metals?
- Are there any existing solutions?
- Which one is the best?
- What is (are) the main limitation(s) regarding the literature review?
- How do you hope to improve or contribute to this?
Moreover, we could not see in the manuscript some basic experiences such as the effect of adsorbent dosage and the effect of temperatures (thermodynamic studies).
Please revise the presentation of the manuscript by using the classic IMRAD (Introduction, Methods, Results and Discussion) format.
Specific aspects:
Please find in the following lines some specific comments.
- Introduction
Revise the introduction as noted in “General aspects”, by highlighting the specific research question and the basic limitations as given in previous literature.
Response: We have considered the Referee's suggestions. The paper has been restructured to include complete and detailed information on the binary removal of Pb and Cd. Particularly, in the Introduction we have addressed the importance and problems of the metallic-removal, using biosorbents, in multimetallic aqueous solutions.
Comments:
Results and discussion
1) Please describe the bands positions with the wavenumber instead of using the positions 1, 2, etc.
Response: It has been changed.
2) Moreover, to better appreciate the displacement, we advise the authors to assign the wavenumber to the corresponding bands which are in the figure.
Response: It has been changed
3) Please write Figure 3a and Figure 3b instead of Fig 3 left and Fig. 3 right.
Response: It has been changed.
4) Figure 3 represents the FTIR spectra of treated OFICM after Pb-loaded or Cd-loaded. This figure corresponds to single removal of Pb(II) or Cd(II). Since the authors are investigating both forms of removal, single and binary removals, they should compare the FTIR spectra of OFICM in both forms of removal. In other word, they should represent the FTIR spectra of OFICM + Pb + Cd and compare the spectra with the previous ones related to the single removals (OFICM + Pb, OFICM + Cd). This comparison could help in the interpretation of the type of interference which occur in the adsorption process.
Response: It has been considered
5) Idem for Figure 4 with the SEM images: SEM images of OFICM + Pb or OFICM + Cd to be compared with the SEM images of OFICM + Pb + Cd.
Response: It has been considered.
6) What can we conclude after analyzing the results obtained in 4) and 5)?
Response: The analysis and interpretation of SEM/EDX have been considered in this revised paper.
7) Influence of pH solution: Figure 5
Please revise the unit of the ordinate axis: the unit of must be in mg/g and not mg/L
Response: It has been changed.
Comments:
The pHPZC was determined to be 4.8 according to Figure 1. As a result:
- for pH < 4.8, OFICM surface is positively charged,
- for pH > 4.8, OFCIM surface is negatively charged.
The authors investigated the influence of the pH in the range from 2 to 7.0. According to the concept of pHPZC, since the OFICM surface is positively charged in the pH range from 2.0 to 4.8 and Pb(II) being a positive free ion, a significant adsorption is not expected. However, in figure 5 for Pb(II) single removal, we have already at a pH value of 2, a significant adsorption of Pb(II) (which is greater than 70 mg/g). This is really contradictory to the concept of pHPZC.
We encourage the authors to refer to the Pourbaix-Diagram of Pb and Cd in order to state the exact state of the two ions and explain what exactly happens. Since electrostatic interactions are not favored, there must be other processes which could explain why at these values, the authors obtained such a high value of.
As a matter of fact, since the pHPZC value is 4.8, what could explain that at a pH value of 4.5 (a value smaller than the 4.8), a maximum value is reached for ? At this value, the surface is still positively charged and Cd(II) and Pb(II) are also positively charged and are not expected to significantly interact with a positive surface charge. This is contradictory with the concept of pHPZC. The authors should really revise their interpretation.
Response: We thank the reviewer for calling our attention. We have modified the pHPZC interpretation, according with our results.
Comment:
In this paper, the authors based their analysis on both forms of removal, single and removal, through OFICM. It is then expected that all results which are given in the paper highlight what occurs in both single and binary removal. In figure 5, the influence of the pH solution was carried out only for the single removal of Pb(II) or Cd(II). In figure 6 for example, the influence of initial metal concentration was carried out for both single and binary removal. It is important to follow the same logic as in figure 6 and to compare the results obtained in both systems. What would be the influence of pH if the removal were carried out following the binary system? As a matter of fact, such binary removal was carried out in the paper of reference [10]. We advise the authors to do the same and to compare the results for both single and binary removal.
Response: It has been considered, including new experiments
Comment:
8) Influence of initial concentration
Lines 161-163: “It is also important to note that the qe values of Pb(II) and Cd(II) in a binary system are lower, particularly for Cd(II), than the corresponding values in a single system. These results indicate that the presence of both metals produce mutual interference effects on their adsorption processes.”
Question: What type of mutual interference effects? Are these effects supported by relevant literature of experience? It would have been better to carry out additional experiences (FTIR, XRD…) to exactly state what is really occurring in this binary system?
Response: The inhibition effect of Pb(II) on co-cation Cd(II) has been considered and treated in the revised paper.
Comment:
9) As already said, the authors should carry out the following experiences: the effect of the adsorbent dosage, effect of temperature (thermodynamic studies). See for example references [10, 23] where these experiences have been carried out.
Response: The referee´s suggestions have been taking into account
Comment:
10) Comparative table
In the comparative table, comparison has been made with mixed systems: those which are treated with NaOH and those which are not. We think that such comparison can only be made with similar systems, we mean systems which have been chemically treated. This table must be revised and compare only systems which have been treated. Untreated OFIC could not be compared with treated OFIC. As a matter of fact, we refer to the investigations which have been carried out in reference [10] by using untreated OFIC (Opuntia ficus indica cladodes). Although the OFIC has not been treated, significant amount of qe was reached: 98.62 mg.g-1 and 30.42 mg.g-1 for Pb(II) and Cd(II) respectively, in a binary removal.
Response: In revised Table 3, we have considered only treated samples.
Comments:
11) Since the authors have conducted single removal, the question arises as to Pb2+ adsorption mechanisms when using OFICM: surface complexation, ion exchange mechanism, etc…However, nothing was said on such possible mechanisms.
12) Miscellaneous
The whole paper contains some sentences which must be corrected (grammar, vocabulary).
Response: It has been duly revised
Comment:
What is the difference between lines “256-260” and lines “261-265”?
It seems to be the same sentences.
Response: The sentence has been removed.
Comment:
General conclusion of the report:
The manuscript must be deeply revised and we recommend to carry out additional experiences. The authors should define the main research problem, recall past results and their limitations. From these limitations, they should define what they intend to do to solve the research question.
Response: We thank the reviewer for his suggestions that have been taken into account.

Reviewer 2 Report
The paper reports on sorption of Cd and Pb ions from single and binary solutions, using chemically modified biomasses. The work seems to be done accurately, and the results are interesting and worth publishing.
Several improvements are necessary.
1. Line 226. Reference Moreira et al (2019) is 34 (in the reference list) instead of 35. Possibly, all references and their numbering should be checked through the manuscript.
2. The hypotheses leading to the "Extended" and "Modified" Langmuir models should be explained briefly.
3. In Figure 9, the extended Langmuir model does not fit at all the Cd sorption capacity data. The analysis should be checked, as most probably some error happened during fitting.
4. The reason of choosing nitrate salts should be explained.
Author Response
RESPONSE TO REVIEWER COMMENTS
- Regarding to Reviewer #2:
Comment: The paper reports on sorption of Cd and Pb ions from single and binary solutions, using chemically modified biomasses. The work seems to be done accurately, and the results are interesting and worth publishing.
Several improvements are necessary.
- Line 226. Reference Moreira et al (2019) is 34 (in the reference list) instead of 35. Possibly, all references and their numbering should be checked through the manuscript.
Response: It has been checked.
Comment:
- The hypotheses leading to the "Extended" and "Modified" Langmuir models should be explained briefly.
Response: It has been explained in lines 909 and 910
Comment:
- In Figure 9, the extended Langmuir model does not fit at all the Cd sorption capacity data. The analysis should be checked, as most probably some error happened during fitting.
Response: It has been checked.
Comment:
- The reason of choosing nitrate salts should be explained.
Response: The lead and cadmium salts used were Pb(NO3)2 and/or Cd(NO3)2·4H2O. These salts are highly soluble in water and are commonly used in this type of works.

Round 2
Reviewer 1 Report
General aspects:
In the first review, some concerns were addressed in link with the main research question. We found that the authors have now addressed this question in the revised manuscript. In the introduction section, the authors showed that the research question is based on a decrease in the adsorption removal in multimetallic systems. Such decrease could be attributed to the competition among metallic ions for accessible binding sites on the biosorbents. Main references have been provided to support the research question. Moreover, some basic experiences such as the effect of adsorbent dosage and the effect of temperatures (thermodynamic studies) have been carried out, and these data give an additional value to the manuscript.
In general, the authors have addressed the main recommendations. However, the paper still contains some sentences which must be corrected.
Another recommendation was to revise the presentation of the manuscript by using the classic IMRAD (Introduction, Methods, Results and Discussion) format. But, this recommendation has not been taken into account
General conclusion of the report:
Since the manuscript has been deeply revised and additional experiences have been carried out, we recommend the manuscript for a publication in the Journal Molecules. We encourage the authors to make a deep proofreading of their manuscript.
The manuscript needs a deep proofreading before a publication. At this step, there is no need to send me back the proofread version of the manuscript.